# Evaluation of the Effectiveness of Boron Neutron Capture Therapy with Iodophenyl-Conjugated *closo*-Dodecaborate on a Rat Brain Tumor Model

**DOI:** 10.3390/biology12091240

**Published:** 2023-09-15

**Authors:** Yoshiki Fujikawa, Yusuke Fukuo, Kai Nishimura, Kohei Tsujino, Hideki Kashiwagi, Ryo Hiramatsu, Naosuke Nonoguchi, Motomasa Furuse, Toshihiro Takami, Naonori Hu, Shin-Ichi Miyatake, Takushi Takata, Hiroki Tanaka, Tsubasa Watanabe, Minoru Suzuki, Shinji Kawabata, Hiroyuki Nakamura, Masahiko Wanibuchi

**Affiliations:** 1Department of Neurosurgery, Osaka Medical and Pharmaceutical University, Osaka 569-8686, Japan; yoshiki.fujikawa@ompu.ac.jp (Y.F.); yusuke.fukuo@ompu.ac.jp (Y.F.); kouhei.tsujino@ompu.ac.jp (K.T.); hideki.kashiwagi@ompu.ac.jp (H.K.); ryo.hiramatsu@ompu.ac.jp (R.H.); naosuke.nonoguchi@ompu.ac.jp (N.N.); motomasa.furuse@ompu.ac.jp (M.F.); toshihiro.takami@ompu.ac.jp (T.T.); wanibuchi@ompu.ac.jp (M.W.); 2Laboratory for Chemistry and Life Science, Institute of Innovative Research, Tokyo Institute of Technology, Yokohama 226-8503, Japan; nishimura.k.am@m.titech.ac.jp (K.N.); hiro@res.titech.ac.jp (H.N.); 3Kansai BNCT Medical Center, Osaka Medical and Pharmaceutical University, Osaka 569-8686, Japan; naonori.ko@ompu.ac.jp (N.H.); shinichi.miyatake@ompu.ac.jp (S.-I.M.); 4Institute for Integrated Radiation and Nuclear Science, Kyoto University, Osaka 590-0494, Japan; takata.takushi.6x@kyoto-u.ac.jp (T.T.); tanaka.hiroki.3e@kyoto-u.ac.jp (H.T.); watanabe.tsubasa.8x@kyoto-u.ac.jp (T.W.); suzuki.minoru.3x@kyoto-u.ac.jp (M.S.)

**Keywords:** boron neutron capture therapy (BNCT), high-grade glioma, boron compound, *closo*-dodecaborate, radiological experimental studies

## Abstract

**Simple Summary:**

In this study, we explored new methods for treating aggressive brain tumors and, particularly, high-grade gliomas. These types of tumors are notoriously difficult to treat effectively; therefore, we focused on a procedure called boron neutron capture therapy (BNCT). BNCT is a particle beam therapy that can selectively destroy tumor cells. However, boronophenylalanine (BPA), a boron compound currently used in this therapy, has limitations. We tested a new compound, boron-conjugated 4-iodophenylbutanamide (BC-IP) and found that even though it does not carry as much boron into the cells as BPA, it remains in the cells longer, which is beneficial for treatment. We also tested BNCT mediated by BC-IP in rats with gliomas and found that it prolonged survival compared to controls (not treated animals and neutrons only). Our work could potentially improve the treatment options for patients with high-grade gliomas. However, further research is required to understand how to best use this new compound for patient benefits.

**Abstract:**

High-grade gliomas present a significant challenge in neuro-oncology because of their aggressive nature and resistance to current therapies. Boron neutron capture therapy (BNCT) is a potential treatment method; however, the boron used by the carrier compounds—such as 4-borono-L-phenylalanine (L-BPA)—have limitations. This study evaluated the use of boron-conjugated 4-iodophenylbutanamide (BC-IP), a novel boron compound in BNCT, for the treatment of glioma. Using in vitro drug exposure experiments and in vivo studies, we compared BC-IP and BPA, with a focus on boron uptake and retention characteristics. The results showed that although BC-IP had a lower boron uptake than BPA, it exhibited superior retention. Furthermore, despite lower boron accumulation in tumors, BNCT mediated by BC-IP showed significant survival improvement in glioma-bearing rats compared to controls (not treated animals and neutrons only). These results suggest that BC-IP, with its unique properties, may be an alternative boron carrier for BNCT. Further research is required to optimize this potential treatment modality, which could significantly contribute to advancing the treatment of high-grade gliomas.

## 1. Introduction

Boron neutron capture therapy (BNCT) is an innovative noninvasive radiotherapy approach for the treatment of cancers such as high-grade gliomas that utilizes the nuclear reaction of the stable isotope boron-10 (^10^B) and thermal neutrons to generate high-linear-energy particles [1,2,3,4]. The approved accelerator-based BNCT for head and neck cancers in Japan has undergone significant advancement [5]; however, current treatments such as BNCT mediated by 4-borono-L-phenylalanine (L-BPA) are not effective for all patients because of the variability in the expression levels of L-type amino acid transporter 1 (LAT-1), which is responsible for BPA uptake into cancer cells [6].

The limitations associated with the current boron carriers approved for use in humans, such as BPA and sodium borocaptate (BSH), necessitate the development of novel carriers. Our previous studies introduced a maleimide-functionalized *closo*-dodecaborate sodium form (MID) and its albumin conjugate (MID-AC) [7,8]. We have previously demonstrated that MID-AC selectively accumulates in tumors and BNCT mediated by MID-AC significantly inhibits tumor growth in a mouse colon tumor model [7]. Importantly, our results revealed the potential of MID-AC to effectively deliver boron for BNCT to treat high-grade gliomas because of its prolonged retention in tumors and anti-tumor effect [9]. BNCT mediated by MID-bovine serum albumin (BSA), another albumin-related boron carrier, also showed long-term retention and anti-tumor effect in a hamster oral cancer model [10]. However, the application of MID-AC is limited by the requirement of high total injection doses [9].

In the continued pursuit of enhancing the therapeutic potential of BNCT and reducing the total injection dose, attention has been focused on the use of small-molecule albumin ligands [11]. Our strategy is designed to form an albumin-boron conjugate post-injection in the blood, rather than preparing it pre-administration. In this context, boron-conjugated 4-iodophenylbutanamide (BC-IP) is a promising candidate [12].

This study aimed to explore the use of BC-IP as a boron carrier in BNCT and its potential impact on the treatment of high-grade gliomas. This underscores the commitment of this research to real-world applications and efficacy, moving beyond laboratory settings and animal models to provide meaningful insights relevant to clinical oncology. The insights gained from this study will significantly advance our understanding of the clinical translation of BNCT.

## 2. Materials and Methods

### 2.1. Boron Compounds

BC-IP with a molecular weight (MW) of 563 was successfully synthesized according to a previous study [12]. Briefly, the synthesis of BC-IP involved a series of reactions, starting with *closo*-dodecaborate, which was transformed into a 1,4-dioxane-closo-dodecaborate complex. This was further modified by amination and iodination to yield water-soluble BC-IP. BPA has extremely low water solubility (0.6–0.7 g/L) [13], whereas BC-IP is water-soluble (at least 14 g/L). The solubility of this compound is a crucial characteristic of potential clinical boron carriers. The chemical structure of BC-IP is shown in Figure 1.

The L-BPA was provided by Interpharma Praha (Prague, Czech Republic). Subsequently, BPA-fructose complex was prepared [14]. Notably, all compounds used in this study were enriched with boron-10.

### 2.2. Cell Culture

The rat glioma cell lines F98 and C6 were used in this study. These cells were respectively obtained from Dr. Rolf Barth at Ohio State University (Columbus, OH, USA) and the Japan Collection of Research Bioresources (JCRB) Cell Bank (National Institute of Biomedical Innovation, Osaka, Japan). All cell cultures were maintained in a medium formulation commonly used in our laboratory, as previously described [9,15,16]. This medium was based on Dulbecco’s modified Eagle’s medium (DMEM), supplemented with 10% fetal bovine serum (FBS), the antibiotics penicillin and streptomycin, and the antifungal agent amphotericin B. The culture conditions were maintained at 37 °C within an atmosphere comprising 5% CO_2_. All necessary components for the formulation of the culture medium were sourced from Gibco Invitrogen (Grand Island, NY, USA).

### 2.3. In Vitro Cellular Uptake of Boron

Intracellular uptake of boron was evaluated using F98 and C6 glioma cells. A total of 4 × 10^5^ cells were plated in a 100-mm dish (Becton, Dickinson, and Company, Franklin Lakes, NJ, USA) and cultured for three days until they reached near-confluency. The culture medium was then replaced with fresh medium containing either 10 µg B/mL of BPA or BC-IP. The cells were subsequently incubated for an additional 3, 6, and 24 h at 37 °C. For boron retention studies, the cells were treated with 10 µg B/mL of BPA or BC-IP for 24 h, after which the medium was replaced with a boron-free medium and incubated for further 1, 3, 6, or 24 h. Following incubation, the cells were rinsed twice with cold phosphate-buffered saline (PBS). Subsequently, we detached the cells using trypsin-ethylenediamine tetra-acetic acid solution. The cells were collected by centrifugation at 200× *g* for 5 min and washed with cold PBS. This procedure was performed twice in order to remove boron attached to the cell membrane. The cells were then digested with 1 N nitric acid solution (FUJIFILM Wako Pure Chemical Industries, Osaka, Japan) overnight at room temperature. The boron concentration of the cell lysate was measured via inductively coupled plasma atomic emission spectroscopy (ICP-AES) (iCAP6300 emission spectrometer, Hitachi High-Technologies, Tokyo, Japan). The boron concentration within the cells was expressed as micrograms of boron (B) per 10^9^ cells.

### 2.4. Rat Glioma Model Using F98 Cells

This study used male Fischer rats, each approximately 10 weeks old and weighing between 200 and 250 g. The rats were rendered unconscious through intraperitoneal administration of a cocktail of anesthetics: medetomidine (0.4 mg/kg; ZENOAQ, Fukushima, Japan), midazolam (2.0 mg/kg; SANDOZ, Yamagata, Japan), and butorphanol (5.0 mg/kg; Meiji Seika, Tokyo, Japan). The head of each rat was secured using a stereotactic frame (Model 900; David Kopf Instruments, Tujunga, CA, USA). After a head skin incision was made using a scalpel, a 1-mm burr hole was made 1 mm posterior to the bregma and 4 mm to the right lateral side. Subsequently, F98 rat glioma cells were implanted into the right brains of these rats. For therapeutic trials, a suspension of 10^3^ F98 rat glioma cells in 10 µL DMEM containing 1.4% agarose (Wako Pure Chemical Industries, Osaka, Japan) was used. For biodistribution investigations, 10^5^ F98 rat glioma cells were injected. All injections were conducted at a steady rate of 20 µL/min and regulated by an automated infusion pump. F98 rat glioma cells were used because they extensively invade normal brains with islands of tumor cells from the tumor mass and are poorly responsive to photon irradiation [17,18]. These surgical procedures have been established and regularly used by our research team in previous studies [9,15,16].

All procedures involving animals adhered to “the Guide for the Care and Use of Laboratory Animals” and were approved by two distinct ethical committees: the Animal Use Review Board of Osaka Medical and Pharmaceutical University (Approval No. 2019-029) and the Ethical Committee of the Institute for Integrated Radiation and Nuclear Science at Kyoto University (KURNS; Kumatori, Osaka, Japan) (Approval No. 2019-9).

### 2.5. Biodistribution Study of Boron Agent in F98 Glioma Model

To assess the biodistribution of boron, the rats were implanted with 10^5^ F98 glioma cells and treated with BC-IP at doses of 5, 10, or 20 mg boron (B) per kg body weight (b.w.) approximately 12–14 days post-implantation, a time window anticipated to allow tumor growth [9,15,16]. The animals were subsequently euthanized at predetermined time points (3, 6, or 24 h after intravenous (i.v.) BC-IP administration) and various tissue samples—namely tumor, brain, blood—were harvested and weighed. Each collected tissue sample was then processed by digestion in a 1N nitric acid solution for 1 week, followed by boron quantification using ICP-AES. The boron concentrations obtained from these assays are expressed as micrograms of boron (B) per gram of tissue.

### 2.6. Survival Analysis of Neutron Irradiation Study for Rat Glioma Models

Survival analysis of in vivo neutron irradiation experiments was performed 15 days after implantation of 10^3^ F98 glioma cells. This pilot study involved a total of 20 rats with F98 glioma, which were randomly assigned into four groups: those left untreated (control, n = 5), those that received neutron irradiation only (irradiation only, n = 2), those that received neutron irradiation following 2 h after i.v. BPA (12 mg B/kg) administration (BNCT with BPA, n = 5) [16], and those subjected to neutron irradiation 3 h after i.v. BC-IP (20 mg B/kg) administration (BNCT with BC-IP, n = 8). This administration-to-irradiation interval was determined based on the results of biodistribution studies. To ensure experimental consistency, all rats were anesthetized prior to the procedures, and either BPA or BC-IP was administered to the relevant groups according to the study design. To concentrate the neutron irradiation exclusively on their heads, all rat bodies were shielded with a plate lined with ^6^LiF ceramic tiles to minimize extraneous exposure. Neutron irradiation for each group was then carried out at the Heavy Water Irradiation Facility at KURNS, at a reactor power of 5 MW with a fixed neutron flux of 3.5 × 10^9^ neutrons/cm^2^/s for a duration of 20 min. After irradiation, all rats were monitored until death or euthanasia. The therapeutic effect of the different treatments was evaluated using Kaplan–Meier survival curves, and the percent increase in life span (%ILS) was computed using the following formula: [(Median Survival Time (MST) of each BNCT group − MST of untreated control group) × 100]/MST of the untreated control group.

### 2.7. Assessing Physical Absorbed Dosages and Compound Biological Effectiveness in In Vivo Neutron Irradiation

The physically absorbed dose—accounting for thermal, epithermal, and fast neutrons as well as gamma rays—was calculated using the formula D_B_ + D_N_ + D_H_ + D_γ_. Here, each component corresponds to the different capture reactions: ^10^B(n, α)^7^Li (D_B_), ^14^N(n, p)^14^C (D_N_), and ^1^H(n, n)^1^H (D_H_), along with gamma ray (D_γ_). Thermal neutron fluence was determined from the radioactivity of gold foil (50 µm thick, 3 mm diameter) attached to the surface of the irradiated rat heads. Because the gold foil is very thin, the dose perturbation effect from the gold foil was calculated to be negligible. Gamma-ray dose was also calculated by thermoluminescence dosimeter attached to the surface of the irradiated rat heads. The doses absorbed to the brain and the tumor in the F98 rat glioma model during BNCT (using BPA and BC-IP) were calculated using data from biodistribution experiments. The estimated photon-equivalent dose (Gy-Eq) was calculated using the equation D_B_ × compound biological effectiveness (CBE) + D_N_ × relative biological effectiveness of nitrogen (RBE_N_) + D_H_ × relative biological effectiveness of hydrogen (RBE_H_) + D_γ_ [19], where RBE_N_ and RBE_H_ are both assumed to be 3.0 [5,20]. The CBEs for normal brain tissue and brain tumor tissue treated with BPA were determined to be 1.35 and 3.8, respectively [20,21,22]. Since BC-IP is quite similar in structure to MID-AC, the CBE of BC-IP was assumed to be 13.4, the same as MID-AC [9].

### 2.8. Statistical Analysis

During the in vitro cellular uptake studies, boron concentrations within different cell lines were analyzed using Student’s *t*-test. Kaplan–Meier curves were used to analyze the survival times. Any significant variance among the groups was determined using log-rank tests. Statistical significance was set at *p* < 0.05 (and deemed significant). All data were processed using JMP^®^ Pro 16.2.0 software (SAS Institute, Cary, NC, USA).

## 3. Results

### 3.1. In Vitro Cellular Uptake of Boron

In this study, we used two types of boron carriers—BPA and BC-IP—on F98 and C6 glioma cells. The boron concentrations in both cell types after exposure to either boron carrier from 3 to 24 h showed an increasing trend over time for both BPA and BC-IP. From 3 to 24 h, BPA showed a significantly higher boron concentration in both cell lines compared to BC-IP, except in C6 glioma cells at 3 h (*p* < 0.05, Student’s *t*-test). However, in terms of boron retention, from 24 + 1 h to 24 + 24 h, the boron retention rate was significantly higher in BC-IP treated cells than in the BPA-treated cells (*p* < 0.05, according to Student’s *t*-test). After switching to a boron-free medium, the boron retention rates for BPA and BC-IP after an additional 1 h of incubation were 35.6% and 81.0% in F98 glioma cells and 40.5% and 63.0% in C6 glioma cells, respectively. After 6 and 24 h of incubation, these rates were 22.8% and 75.6% (6 h) and 16.3% and 32.5% (24 h) in F98 glioma cells, and 23.6% and 54.3% (6 h) and 14.0% and 19.1% (24 h) in C6 glioma cells, respectively. When switched to a boron-free medium, boron retention in BPA-treated cells decreased rapidly, whereas BC-IP-treated cells showed a gradual decrease in boron retention over time. The results are shown in Figure 2.

### 3.2. Biodistribution Study of the Boron Agent in F98 Glioma Model

Boron concentrations in the tumors, normal brain tissue, and blood of F98 glioma-bearing rats were measured 3 h after iv administration of BC-IP at 20, 10, and 5 mg B/kg (Figure 3a). In practice, the dosage was adjusted by changing the concentration. In addition, BC-IP was administered at 10 mg B/kg, and then boron concentrations were measured 3, 6, and 24 h later (Figure 3b), based on our previous studies [9]. The results are also shown in Table 1. In the former (Figure 3a), the boron concentration increased in a concentration-dependent manner, but the boron concentration was low in the tumor. The tumor-to-normal brain ratios at 20, 10, and 5 mg B/kg at 3 h were found to be 3.3, 6.4, and 2.1, respectively, while the tumor-to-blood ratios at 3 h were 0.5, 1.04, and 0.6, respectively. For the latter (Figure 3b), boron concentrations were highest 3 h after BC-IP administration and then decreased over time.

### 3.3. In Vivo Survival Analysis of the Neutron Irradiation Study

BC-IP was administered 3 h before irradiation because that time-point was the highest boron concentration in the tumor in the biodistribution study. The MSTs for each group are as follows: control group, 24 days [95% Confidence Interval (CI), 24–32 days]; irradiation-only group, 28.5 days [95% CI, 28–29 days]; BNCT with BPA group, 39 days [95% CI, 34–49 days]; and BNCT with BC-IP group, 34 days [95% CI, 28–37 days].

Although the BNCT with BC-IP group had a significantly shorter survival extension than the BNCT with BPA group (*p* = 0.016, according to the log-rank test), they had a significantly longer survival extension than the untreated control and irradiation-only group (compared to the control group, *p* = 0.0086; compared to the irradiation-only group, *p* = 0.019, respectively, log-rank test) (Figure 4).

The %ILS values were as follows: BNCT with BC-IP, 41.7%, and BNCT with BPA, 62.5%. A detailed description of these results is presented in Table 2. No radiotoxicity was observed in any of the groups.

### 3.4. Assessing Physical Dosages and Compound Biological Effectiveness of In Vivo Neutron Irradiation

The physical and photon-equivalent doses delivered to the brains and tumors during neutron irradiation experiments were estimated using the CBE factor for each boron carrier, which was derived from reference data [9]. In BPA-BNCT at 2 h, this estimation also considered the average tumor boron concentrations (Tumor 17.8, Brain 4.3 μg B/g) obtained from in vivo biodistribution studies [16]. The estimated photon-equivalent doses delivered to the tumor during BPA-BNCT at 2 h was 13.7 Gy-Eq. In contrast, for BNCT with BC-IP at 3 h, the doses were 4.6 Gy-Eq (Table 3).

## 4. Discussion

This study focused on the application of BC-IP, a novel boron carrier, in BNCT for the treatment of high-grade gliomas. Through in vivo studies, including neutron irradiation, we observed favorable results, highlighting the potential of BC-IP to enhance the therapeutic potential of BNCT. 

Human serum albumin (HSA) is the most abundant polypeptide protein in blood plasma. HSA, which weighs 66.5 kDa and consists of 585 amino acids, is present in the plasma at concentrations of 35–50 g/L [23,24,25]. It has many important physiological functions. For example, it significantly contributes to the regulation of plasma oncotic pressure and fluid distribution. It also functions as a transport molecule for various internal and external substances due to its remarkable ligand-binding capacity [26]. In addition, HSA accumulates in malignant tumor and inflammatory tissues because of the enhanced permeability and retention (EPR) effect [27,28]. Furthermore, rapidly growing tumor cells take up significant amounts of extracellular proteins, including serum albumin, which represents the total volume of extracellular proteins in the tumor cytoplasm. This is due to the fact that serum albumin serves as an important nutrient source for tumors to grow [29,30]. Hence, HSA has been studied as a potential drug delivery system. As an example, nab-paclitaxel—an albumin–paclitaxel nanoparticles complex—was first approved in the United States for the treatment of metastatic breast cancer in 2005 [31,32]. Thereafter, the FDA expanded approval to other type of cancers, such as non-small cell lung cancer and pancreatic cancer [33,34]. Upon intravenous administration, nab-paclitaxel transforms into soluble albumin-bound complexes that accumulate in tumors via mechanisms such as transcytosis and the EPR effect [35]. Some albumin–drug conjugates are currently under clinical study, including the methotrexate–albumin conjugate (MTX-HSA) and the doxorubicin maleimide derivative (INNO-206) [36,37,38,39,40,41,42]. Considering these factors, we believe that albumin can serve as an effective boron carrier for BNCT. Accordingly, we have been engaged in the development and research of albumin-related drugs for BNCT in recent years [7,9,12,15].

Recently, we reported that BNCT with MID-AC was effective in the F98 glioma rat model [9]. This study demonstrated that the drug remained in the tumor for an extended period, suggesting its potential for a stable and sustained boron capture reaction through neutron irradiation. However, a challenge with this drug is its increased molecular weight due to the conjugation of albumin during the synthesis stage, leading to the requirement for a larger dosage. For the MID albumin conjugates, a boron dose of 7.5 mg/kg body weight was equivalent to a total dose of 420 mg/kg body weight [41,43]. To address this issue, we developed BC-IP, a compound that can bind to albumin in the blood after intravenous administration [12].

In the in vitro drug exposure experiments, BC-IP exhibited significantly lower intracellular boron uptake than BPA, with only a slight increase in uptake as the exposure time was extended (Figure 2a,b). In contrast, in terms of clearance, a large proportion of BPA was expelled one hour after exposure, whereas BC-IP displayed superior retention (Figure 2c,d). Because BC-IP was originally designed to bind with albumin in the blood after intravenous administration, its precise behavior in this in vitro experiment is unclear. Although the boron concentration in BC-IP was lower than that in BPA, this may be attributed to the unique characteristics of the drug. However, the high retention rate of BC-IP indicates its potential suitability as a boron carrier for BNCT. This property is particularly significant, considering the current limitations of BPA, the sole boron compound approved for clinical use. Our in vitro results indicated that BPA was rapidly eliminated from the blood upon cessation of administration. Other studies also showed that BPA was quickly cleared after exposure was terminated [44,45]. Therefore, in current clinical BNCT, patients must be continuously administered large amounts of BPA to maintain the necessary concentration of boron in the blood throughout neutron irradiation [46]. The superior retention rate of BC-IP could alleviate this issue, indicating its potential as a novel boron carrier for BNCT.

In the in vivo biodistribution study, we evaluated boron concentrations in the tumor, normal brain, and blood of F98 glioma-bearing rats following the intravenous administration of the boron compound BC-IP at various concentrations. The results showed that the boron concentration increased with the administered dose. In addition, boron concentrations peaked at 3 h after BC-IP administration and gradually decreased over time. Therefore, we performed our in vivo neutron irradiation experiment 3 h after the administration of BC-IP 20 mg B/kg. However, boron accumulation in the tumor was quite low, which may explain the shorter survival time in the irradiation experiment compared to the BNCT with BPA group. By time, boron accumulation in tumors decreased over time, and boron accumulation was almost completely cleared 24 h after administration; however, the T/Br ratio was high at both 3 and 6 h after administration. Although boron accumulation needs to be improved, based on the in vitro boron clearance results, irradiation at 6 h after BC-IP administration may provide a survival benefit equivalent to 3 h after administration because of the relatively high retention rate up to the 6 h time point. The high retention rate may avoid the need for a continuous infusion of boron compounds, as mentioned earlier, and may allow for a variety of irradiation plans such as adjusting the irradiation time and flexible scheduling.

In our in vivo neutron irradiation study, although the prolonged survival in the BC-IP group that received BNCT was not as favorable as that in the BPA group, it was still significantly longer than the prolonged survival observed in both the untreated control and irradiation-only groups. It is worth noting that although boron accumulation in the tumor was extremely low in BC-IP in our biodistribution study, the results were better than those of the control groups. This result may suggest that the biological effect of BC-IP is relatively high. Although the factors associated with CBE are complex, not fully understood, and still controversial, we calculated them as reference values for easy comparison with previous reports.

BPA enters the tumor mainly via LAT-1 [6], whereas the novel compound BC-IP enters the tumor via the EPR effect [28]. In addition, boron uptake when conjugated to albumin could be attributed not only to the EPR effect, but also to an increase in catabolism by the tumor or through other albumin-specific transporters such as gp60 and secreted protein acidic and rich in cysteine (SPARC) [35,47]. In other words, these agents are taken up through different mechanisms. Therefore, BC-IP alone is not sufficiently effective. However, when used in combination with BPA, it may contribute to solving the problem of BPA-negative cells. 

However, glioma cells infiltrate the normal brain parenchyma where the blood–brain barrier (BBB) remains intact. Therefore, the delivery of these agents remains a major challenge. Our solution to overcome this challenge may be convection-enhanced delivery (CED), which allows for the direct administration of therapeutic agents into the tumor or brain parenchyma. This technique bypasses the BBB using pressure-driven bulk flow, offering significant pharmacokinetic advantages over traditional iv administration [48,49,50]. Therefore, the combination of CED and novel boron carriers such as BC-IP could potentially enhance the effectiveness of BNCT treatment for high-grade gliomas. However, this would require the use of molecular targets or other modifications to BC-IP. Thus, we are currently working to further develop BC-IP and are considering using it in combination with CED or BPA. We suggest that bevacizumab (BEV), an anti-vascular endothelial growth factor (VEGF) antibody, is another option. In clinical practice, the addition of bevacizumab to BNCT has already been shown to prolong survival [51] and is expected to be an effective treatment regimen in the future.

## 5. Conclusions

Our research illustrates the potential of BC-IP as an alternative boron carrier for BNCT, specifically within the context of an F98 glioma model. Despite BC-IP displaying lower intracellular boron uptake in comparison to BPA, it exhibits superior retention, highlighting its potential for future BNCT applications. Continued research is crucial to optimize these methodologies, promising significant advancements in BNCT for glioma treatment.

## Figures and Tables

**Figure 1 biology-12-01240-f001:**
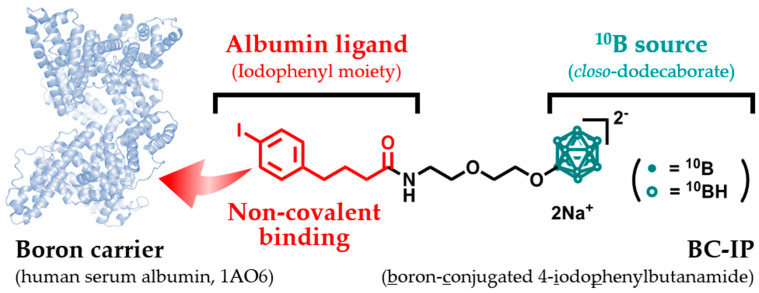
The chemical structure and synthetic scheme of BC-IP.

**Figure 2 biology-12-01240-f002:**
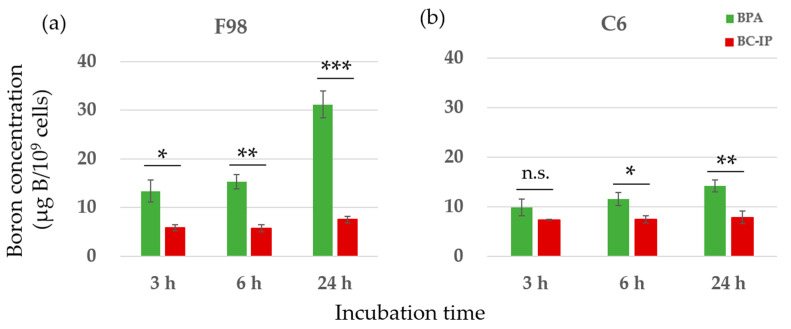
In vitro boron uptake and retention rates. (**a**,**b**) Boron concentrations in F98 and C6. Each cell line was exposed to 10 µg B/mL of BC-IP or BPA for 3, 6, and 24 h. Data is expressed in terms of mean ± standard deviation (SD). The boron concentration of BC-IP from 3 h to 24 h was significantly lower than that of BPA except in C6 at 3 h. n.s., not significant. (**c**,**d**) Each cell line was exposed to 10 µg B/mL BC-IP or BPA for 24 h and subsequently incubated with a boron-free medium for 1, 3, 6, and 24 h. The retention of boron in cells treated with BPA decreased rapidly, while that of cells treated with BC-IP decreased gradually. The boron retention rate of BC-IP from additional 1 to 24 h was significantly higher than that of BPA at the same time-point in both F98 and C6. * *p* < 0.05, ** *p* < 0.01, *** *p* < 0.001, **** *p* < 0.0001.

**Figure 3 biology-12-01240-f003:**
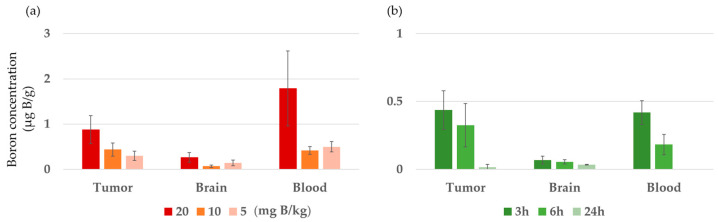
Biodistribution of BC-IP in F98 rat glioma models after intravenous injection. (**a**) Boron concentrations 3 h after administration of each BC-IP dosage. (**b**) Boron concentration by time after administration of BC-IP 10 mg B/kg. Data is expressed in terms of mean ± standard deviation (SD).

**Figure 4 biology-12-01240-f004:**
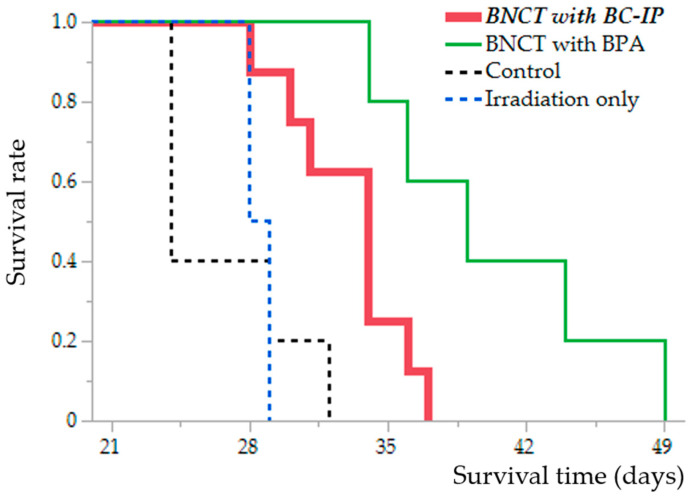
Kaplan–Meier survival curves of the F98 glioma rat models after neutron irradiation. Survival time data, in days post-implantation of F98 rat glioma cells, are illustrated across four distinct groups: Untreated control (black dotted line), Irradiation-only (blue dotted line), BNCT following 2 h of boronophenylalanine (BPA) administration (green line), and BNCT following 3 h of boron-conjugated 4-iodophenylbutanamide (BC-IP) administration (red bold line). Statistically significant differences were found when comparing the survival rates of the BNCT with BC-IP group to the untreated control and irradiation-only group (to untreated control group, *p* = 0.0086; to irradiation-only group, *p* = 0.019, respectively, according to the log-rank test).

**Table 1 biology-12-01240-t001:** Biodistribution of boron in the tumor, brain, and blood after intravenous administration of BC-IP in rat tumor model.

BC-IP Dose (mg B/kg)	Time (h)	n	Boron Concentration ± SD (µg B/g) ^a^			Ratio	
			Tumor	Brain	Blood	T/Br ^b^	T/Bl ^c^
20	3	4	0.9 ± 0.3	0.3 ± 0.1	1.8 ± 0.8	3.3	0.5
10	3	4	0.4 ± 0.1	0.07 ± 0.03	0.4 ± 0.09	6.4	1.04
	6	3	0.3 ± 0.2	0.05 ± 0.02	0.18 ± 0.08	6.0	1.8
	24	3	0.02 ± 0.02	0.03 ± 0.0	0	0.5	-
5	3	4	0.3 ± 0.1	0.1 ± 0.06	0.5 ± 0.1	2.1	0.6

^a^ Data is expressed as mean ± standard deviation (SD). ^b^ T/Br, tumor-to-brain. ^c^ T/Bl, tumor-to-blood.

**Table 2 biology-12-01240-t002:** Survival times of F98 glioma rat models after neutron irradiation.

Group	*n*	Survival Time (Days)			%ILS ^a^	*p*-Value ^b^
		Mean ± SD	Median	95% CI		
Control	5	26.6 ± 3.3	24	24–32		
Irradiation-only	2	28.5 ± 0.5	28.5	28–29	18.8	0.98
BNCT with BPA	5	40.4 ± 5.5	39	34–49	62.5	0.0015
BNCT with BC-IP	8	33 ± 2.9	34	28–37	41.7	0.0086

^a^ Percentage increase in lifespan (%ILS) was determined based on the mean survival time (MST) of the untreated control group. %ILS means {(MST of each BNCT group − MST of untreated control group) × 100}/MST of untreated control group. ^b^ *p*-values were calculated using the log-rank test and compared to the untreated control group based on the results obtained from the Kaplan–Meier curves in the neutron irradiation study for the F98 glioma rat models. SD, standard deviation. CI, confidence interval.

**Table 3 biology-12-01240-t003:** Absorbed and photon-equivalent doses to the brain and tumors in rat tumor models.

Group	Absorbed Dose ^a^ (Gy)		Photon-Equivalent Dose ^b^ (Gy-Eq)	
	Brain	Tumor	Brain	Tumor
Untreated	0.0	0.0	0.0	0.0
Irradiation-only	1.8	1.8	2.7	2.7
BNCT with BPA 2 h	2.5	4.7	3.7	13.7
BNCT with BC-IP 3 h	1.9	1.9	- *	4.6

^a^ The physically absorbed dose is derived from the ^10^B (n, α)^7^Li, ^14^N(n, p)^14^C, and ^1^H(n, n)^1^H reactions induced by thermal, epithermal, and fast neutron fluxes and gamma rays in the irradiating neutrons. It is calculated by the following equation: absorbed dose (Gy) = D_B_ + D_N_ + D_H_ + D_γ_. ^b^ The estimated photon-equivalent dose was calculated using this formula: D_B_ × compound biological effectiveness (CBE) + D_N_ × relative biological effectiveness of nitrogen (RBE_N_) + D_H_ × relative biological effectiveness of hydrogen (RBE_H_) + D_γ_. The value for RBE_N_ and RBE_H_ are set to 3.0. For BPA, the CBE factor for the normal brain tissue was set at 1.35. * In the case of BC-IP, the CBE for normal brain tissue was not determined; therefore, the field was left blank.

## Data Availability

All data analyzed in this study are available upon reasonable request from the corresponding author. The JMP Pro version 16.2.0. software (SAS, Cary, NC, USA) was used for the statistical analysis.

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
