# Peer review of "Evaluation of the Effectiveness of Boron Neutron Capture Therapy with Iodophenyl-Conjugated closo-Dodecaborate on a Rat Brain Tumor Model"

_biology, 2023, doi:10.3390/biology12091240_

Round 1
Reviewer 1 Report
The article by Fujikawa et al. is relevant in the field of BNCT. Currently, the development of new boron compounds and their evaluation in radiobiological models is an actual need, to overcome those limitations associated to BPA and BSH, the two boron compounds approved for their use in humans. Particularly, BPA has been used for a wide range of pathologies and its principal issue is its poor retention in the tumor. This constraint makes necessary a BPA infusion during irradiation, with the possibility of dislodgement of the injection needle in the middle of the irradiation and the need for complicated settings. In this way, new boron compounds that could improve tumor retention are of significant interest.
This article needs revision. Please, could you address the following comments?
Simple summary:
Could you please re-write the following definition?: “In BNCT, a compound containing boron is injected into the body, which is then targeted by a beam of neutrons to destroy tumor cells.” It could be misleading for someone not familiar with BNCT.
Please add “mediated by BNCT” in the phrase: “We also tested BC-IP mediated by BNCT in rats with gliomas...”. Please complete that BC-IP/BNCT “prolonged survival vs controls (not treated animals and neutrons only)”. .
Abstract:
Please add “mediated by BNCT” and “(not treated animals and neutrons only)” in the phrase: "Furthermore, despite lower boron accumulation in tumors, BC-IP mediated by BNCT showed significant survival improvement in glioma-bearing rats compared to controls (not treated animals and neutrons only)".
Keywords:
Please add: “radiobiological experimental studies”
Introduction:
Line 55: Please correct: “however, current treatments such as 4-borono-L-phenylalanine (L-BPA) mediated by BNCT treatments...”
Line 58: “The limitations associated with the current boron carriers approved for their use in humans like BPA and BSH necessitate the development of novel carriers”
Line 61/62: Please add “MID-AC mediated by BNCT” in the following phrase: “We have previously demonstrated that MID-AC selectively accumulates in tumors and MID-AC mediated by BNCT significantly inhibits tumor growth in a mouse colon tumor model [7,9,10]”. Related to this phrase, are references 9 and 10 correct? They do not mention BNCT in the abstract. MID-AC or MID-BSA was previously demonstrated therapeutically useful in glioma and oral cancer animal models, apart from colon tumor model. Please check which references should be mention here.
Line 64/65: In reference 11 the authors did not evaluate MID-AC microdistribution inside the cell to mention that boron neutron capture reactions occurred preferentially in the tumor cell nucleus. Please verify this sentence: “because of its efficient induction of boron neutron capture reactions in the tumor cell nucleus and its prolonged retention in tumor cells”.
Please revise the following phrase: “This study aimed to further explore the use of BC-IP as a boron carrier in BNCT, with a particular focus on in vivo tumor accumulation and its potential impact on the treatment of high-grade gliomas. The main objective of this study was to validate the enhanced therapeutic potential of BNCT combined with BC-IP, an innovative approach that has been extensively explored through in vivo studies using neutron irradiation”. (1) In this study, the authors evaluate in vitro and in vivo boron accumulation. (2) The phrase “that has been extensively explored through in vivo studies using neutron irradiation” seems that this study has been made before, and the authors did not mention any references related to this. I think this phrase has to be reformulated.
Materials and methods:
Section 2.1:
Line 87: Could you please compare BPA vs BC-IP solubility? One of the principal limitations related to BPA is solubility. So perhaps in this study not only the authors overcome the issue of “retention in the tumor” but also “solubility”. It would be interesting if the authors could clarify this aspect.
Is it possible for the authors to add a scheme of the boron compound? I think this could help identifying and suspect how this boron carrier could enter the cells in vitro, without being complexed with albumin. This topic was mentioned in the discussion section.
Section 2.4: why did you choose to implant F98 cells instead of C6? if it is regarding the results obtained, please clarify this.
Section 2.5: Regarding to: “To assess the biodistribution of boron, the rats were implanted with 105 F98 glioma cells 142 and treated with BC-IP at doses of 5, 10, or 20 mg boron (B) per kg body weight (b.w.) 143 approximately 12-14 days post-implantation, a time window anticipated to allow substan- 144 tial tumor growth. The animals were subsequently euthanized at predetermined time 145 points and various tissue samples — namely tumor, brain, blood — were harvested and 146 weighed. Each collected tissue sample was then processed by digestion in a 1N nitric acid 147 solution for 1 week, followed by boron quantification using ICP-AES. The boron concen- 148 trations obtained from these assays are expressed as micrograms of boron (B) per gram of 149 tissue.”
*BC-IP doses were chosen based on previous studies?
*”12-14 days post-implantation, a time window anticipated to allow substantial tumor growth” this is based on previous studies?
*”The animals were subsequently euthanized at predetermined time 145 points”: which ones?
*”Each collected tissue sample was then processed by digestion in a 1N nitric acid solution for 1 week”: tissues were remained in nitric acid during 1 week before boron quantification?
Section 2.6: Regarding to: “Survival analysis of in vivo neutron irradiation experiments was performed 15 days 152 after implantation of 103 F98 glioma cells. The study involved a total of 20 rats with F98 153 glioma, which were randomly assigned into four groups: those left untreated (control, n 154 = 5), those received neutron irradiation only (irradiation only, n = 2), neutron irradiation 155 following 2.5 h after intravenous BPA administration (BNCT with BPA, n = 5), and those 156 subjected to neutron irradiation 3 h after intravenous BC-IP administration (BNCT with 157 BC-IP, n = 8)”.
*I recommend mentioning that this is a pilot study, as number of animals is quite low, principally in the irradiation only group.
* “following 2.5 h after intravenous BPA administration” please add reference.
* “neutron irradiation 3 h after intravenous BC-IP administration” please add a comment relative to the biodistribution studies shown in this work that helped to select this time after injection.
Section 2.7:
In reference 11, Kashiwagi et al. mentioned “no statistically significant differences in the respective MSTs in BNCT groups. Therefore, we assumed the estimated photon-equivalent doses of brain tumor were also equivalent between BNCT using BPA 2.5h and BNCT using MID-AC at 2.5 h or 24 h after the neutron irradiation experiment for F98 rat glioma models” Moreover, the authors of the article in review mentioned that “Since BC-IP is quite similar in structure to MID-AC, the CBE of BC-IP was assumed 183 to be 13.4, the same as MID-AC [11]”.
CBE values depend on boron concentration and microdistribution in tumor and inside the cells (between other aspects). Moreover, calculating Gy-eq with the “traditional formula” lacks information, like cell repairing. Could you please explain why did you reported Gy-Eq doses instead of Gy Absorbed doses?
Results:
*Section 3.1: It is interesting to mention that both cell lines accumulate BC-IP in the same way, rather than BPA. This shows one of the issues mentioned in the introduction, related to the differences in BPA uptake between tumors.
Figure 1 C and D: statistical analyses are lacking.
* Section 3.2: Please revise redaction and unify redaction with M&M the boron doses used for biodistribution studies. This section was difficult to follow.
Moreover, it would be useful for the reader to explain why the authors chose 500 ugB/ml to explore 3, 6, and 24 hours after injection. Is it based on previous studies?
Table 1 and figure 2 shows very low boron concentration values in the tumor. Please explain in the text carefully (it was mentioned this in the discussion section).
*Section 3.3: Why did you choose 3 h after BC-IP injection if blood has the highest boron concentration at this point? Why you did not choose 6 h after injection that has lower boron concetration in blood?
*Boron concentration values used for dose planning corresponding to BPA group are lacking. How did you prescribe the dose in these BNCT in vivo irradiations? Which is your dose-limiting tissue? Did you evaluate radiotoxicity?
*Section 3.4: “The physical and photon-equivalent doses delivered to the brains and tumors during 263 neutron irradiation experiments were estimated using the CBE factor for each boron car- 264 rier, which was derived from reference data [11]. This estimation also considered the av- 265 erage tumor boron concentrations obtained from in vivo biodistribution studies [16]”
Why the authors did not use the values obtained in the biodistribution studies reported in this article? This comment is related to the comment made before related to Gy-Eq calculations.
Discussion:
“This study focused on the application of BC-IP, a novel boron carrier, in BNCT for 279 the treatment of high-grade gliomas. Through extensive in vivo studies, including neutron 280 irradiation, we observed promising results, highlighting the potential of BC-IP to enhance 281 the therapeutic potential of BNCT”.
I recommend to erase the word “extensive”, as I the number of animals used in this study is quite low.
*In the statement “whereas the novel com- 354 pound BC-IP enters the tumor via the EPR effect”.
I recommend to mention “between others” as boron uptake when conjugated with albumin could be also due to an increase in albumin catabolism by the tumor, or other albumin specific transporters like gp60 and SPARC (example Kikuchi et al. 2016 – ref 6).
*Finally, related to your comments on combining BPA + BC-IP and CED protocol, I recommend the authors to analyze and discuss the article by Fukuo et al. 2020 (ref 16) to enrich their future protocols and discuss them in the text.
Author Response
Dear Reviewer 1,
Thank you very much for reviewing our manuscript and offering valuable advice. We have addressed your comments with point-by-point responses and revised the manuscript accordingly. We believe that the comments have made our manuscript deeper and better than before. All co-authors agreed and we have finished the revision. Please check the history of changes in the manuscript WORD review and responses to the comments below.
Simple summary:
Could you please re-write the following definition?: “In BNCT, a compound containing boron is injected into the body, which is then targeted by a beam of neutrons to destroy tumor cells.” It could be misleading for someone not familiar with BNCT.
Ans. We re-wrote definition. Please refer to line 26.
Please add “mediated by BNCT” in the phrase: “We also tested BC-IP mediated by BNCT in rats with gliomas...”. Please complete that BC-IP/BNCT “prolonged survival vs controls (not treated animals and neutrons only)”. .
Ans. We added “mediated by BNCT”. At line 31 and “prolonged survival vs controls (not treated animals and neutrons only)” at line 32.
Abstract:
Please add “mediated by BNCT” and “(not treated animals and neutrons only)” in the phrase: "Furthermore, despite lower boron accumulation in tumors, BC-IP mediated by BNCT showed significant survival improvement in glioma-bearing rats compared to controls (not treated animals and neutrons only)".
Ans. We added “mediated by BNCT” and “(not treated animals and neutrons only)”. Please refer to line 43-44.
Keywords:
Please add: “radiobiological experimental studies”
Ans. We added this keyword at line 49.
Introduction:
Line 55: Please correct: “however, current treatments such as 4-borono-L-phenylalanine (L-BPA) mediated by BNCT treatments...”
Ans. We corrected as you pointed at line 57.
Line 58: “The limitations associated with the current boron carriers approved for their use in humans like BPA and BSH necessitate the development of novel carriers”
Ans. We corrected as you pointed at line 60-61.
Line 61/62: Please add “MID-AC mediated by BNCT” in the following phrase: “We have previously demonstrated that MID-AC selectively accumulates in tumors and MID-AC mediated by BNCT significantly inhibits tumor growth in a mouse colon tumor model [7,9,10]”. Related to this phrase, are references 9 and 10 correct? They do not mention BNCT in the abstract. MID-AC or MID-BSA was previously demonstrated therapeutically useful in glioma and oral cancer animal models, apart from colon tumor model. Please check which references should be mention here.
Ans. We added “MID-AC mediated by BNCT” at line 64. You are right. I am sorry that I took wrong references. I deleted previous 9 and 10 references. Instead, we added MID-BSA reference [10] at line 69-71. Thank you for telling us.
- Monti Hughes, A.; Goldfinger, J.A.; Palmieri, M.A.; Ramos, P.; Santa Cruz, I.S.; De Leo, L.; Garabalino, M.A.; Thorp, S.I.; Curotto, P.; Pozzi, E.C.C.; et al. Boron Neutron Capture Therapy (BNCT) Mediated by Maleimide-Functionalized Closo-Dodecaborate Albumin Conjugates (MID:BSA) for Oral Cancer: Biodistribution Studies and In Vivo BNCT in the Hamster Cheek Pouch Oral Cancer Model. Life (Basel) 2022, 12, doi:10.3390/life12071082.
Line 64/65: In reference 11 the authors did not evaluate MID-AC microdistribution inside the cell to mention that boron neutron capture reactions occurred preferentially in the tumor cell nucleus. Please verify this sentence: “because of its efficient induction of boron neutron capture reactions in the tumor cell nucleus and its prolonged retention in tumor cells”.
Ans. We are sorry for misleading words. We corrected at line 67-69.
because of its prolonged retention in tumors and anti-tumor effect [9]
Please revise the following phrase: “This study aimed to further explore the use of BC-IP as a boron carrier in BNCT, with a particular focus on in vivo tumor accumulation and its potential impact on the treatment of high-grade gliomas. The main objective of this study was to validate the enhanced therapeutic potential of BNCT combined with BC-IP, an innovative approach that has been extensively explored through in vivo studies using neutron irradiation”. (1) In this study, the authors evaluate in vitro and in vivo boron accumulation. (2) The phrase “that has been extensively explored through in vivo studies using neutron irradiation” seems that this study has been made before, and the authors did not mention any references related to this. I think this phrase has to be reformulated.
Ans. We corrected as simple as possible. Please refer to line 78-80.
This study aimed to further explore the use of BC-IP as a boron carrier in BNCT its potential impact on the treatment of high-grade gliomas.
Materials and methods:
Section 2.1:
Line 87: Could you please compare BPA vs BC-IP solubility? One of the principal limitations related to BPA is solubility. So perhaps in this study not only the authors overcome the issue of “retention in the tumor” but also “solubility”. It would be interesting if the authors could clarify this aspect.
Ans. We added water solubility of BPA and BC-IP at line 93-94.
BPA has extremely low water solubility (0.6 - 0.7 g/L), as noted in the Borofalan interview form, whereas BC-IP is water-soluble (at least 14 g/L).
Is it possible for the authors to add a scheme of the boron compound? I think this could help identifying and suspect how this boron carrier could enter the cells in vitro, without being complexed with albumin. This topic was mentioned in the discussion section.
Ans. We presented scheme as Fig A1(page 12, Appendix A). How do you like it?
Section 2.4: why did you choose to implant F98 cells instead of C6? if it is regarding the results obtained, please clarify this.
Ans. F98 cells were used because they extensively invade normal brain and are poorly responsive to radiotherapy. We added new references. Please refer to line at 144-146.
- Barth, R.F. Rat brain tumor models in experimental neuro-oncology: the 9L, C6, T9, F98, RG2 (D74), RT-2 and CNS-1 gliomas. J Neurooncol 1998, 36, 91-102, doi:10.1023/a:1005805203044.
- Barth, R.F.; Kaur, B. Rat brain tumor models in experimental neuro-oncology: the C6, 9L, T9, RG2, F98, BT4C, RT-2 and CNS-1 gliomas. J Neurooncol 2009, 94, 299-312, doi:10.1007/s11060-009-9875-7.
Section 2.5: Regarding to: “To assess the biodistribution of boron, the rats were implanted with 105 F98 glioma cells 142 and treated with BC-IP at doses of 5, 10, or 20 mg boron (B) per kg body weight (b.w.) 143 approximately 12-14 days post-implantation, a time window anticipated to allow substan- 144 tial tumor growth. The animals were subsequently euthanized at predetermined time 145 points and various tissue samples — namely tumor, brain, blood — were harvested and 146 weighed. Each collected tissue sample was then processed by digestion in a 1N nitric acid 147 solution for 1 week, followed by boron quantification using ICP-AES. The boron concen- 148 trations obtained from these assays are expressed as micrograms of boron (B) per gram of 149 tissue.”
*BC-IP doses were chosen based on previous studies?
Ans. Exactly. We previously conducted MID-AC/BNCT study at dose of 20mg B/kg. So first, we started this experiment at quarter dose (5mg B/kg). Then we did half dose (10mg B/kg). At this dose, we also conducted biodistribution at different time-points (3, 6, or 24 h after administration). At 3h point was the highest boron concentration, so we did at 3h only at 20mg B/kg.
*”12-14 days post-implantation, a time window anticipated to allow substantial tumor growth” this is based on previous studies?
Ans. Exactly. This is based on previous our studies. We added our references at line 157. These 10^5 F98 models will become weak and die in about 2 weeks due to brain herniation.
*”The animals were subsequently euthanized at predetermined time 145 points”: which ones?
Ans. This sentence was not enough, so we added the detailed time window (3, 6, or 24h after i.v. BC-IP). Please refer to line 158-159.
*”Each collected tissue sample was then processed by digestion in a 1N nitric acid solution for 1 week”: tissues were remained in nitric acid during 1 week before boron quantification?
Ans. You are right, tissues were remained in nitric acid during 1 week before quantification.
Section 2.6: Regarding to: “Survival analysis of in vivo neutron irradiation experiments was performed 15 days 152 after implantation of 103 F98 glioma cells. The study involved a total of 20 rats with F98 153 glioma, which were randomly assigned into four groups: those left untreated (control, n 154 = 5), those received neutron irradiation only (irradiation only, n = 2), neutron irradiation 155 following 2.5 h after intravenous BPA administration (BNCT with BPA, n = 5), and those 156 subjected to neutron irradiation 3 h after intravenous BC-IP administration (BNCT with 157 BC-IP, n = 8)”.
*I recommend mentioning that this is a pilot study, as number of animals is quite low, principally in the irradiation only group.
Ans. We mentioned that this was a pilot study (Line 166).
* “following 2.5 h after intravenous BPA administration” please add reference.
Ans. We added reference [15] of our previous study (Line 169-170). We are sorry that 2 h is correct, not 2.5 h. We corrected.
* “neutron irradiation 3 h after intravenous BC-IP administration” please add a comment relative to the biodistribution studies shown in this work that helped to select this time after injection.
Ans. We added “This administration-to-irradiation interval was determined based on the results of biodistribution studies” (Line 171-172).
Section 2.7:
In reference 11, Kashiwagi et al. mentioned “no statistically significant differences in the respective MSTs in BNCT groups. Therefore, we assumed the estimated photon-equivalent doses of brain tumor were also equivalent between BNCT using BPA 2.5h and BNCT using MID-AC at 2.5 h or 24 h after the neutron irradiation experiment for F98 rat glioma models” Moreover, the authors of the article in review mentioned that “Since BC-IP is quite similar in structure to MID-AC, the CBE of BC-IP was assumed 183 to be 13.4, the same as MID-AC [11]”.
CBE values depend on boron concentration and microdistribution in tumor and inside the cells (between other aspects). Moreover, calculating Gy-eq with the “traditional formula” lacks information, like cell repairing. Could you please explain why did you reported Gy-Eq doses instead of Gy Absorbed doses?
Ans. As you have pointed out, the notation in Gy-Eq has several problems, and we think that it is highly debatable whether it really shows an equivalent biological effect to Gy, including the calculation method used in this manuscript. With regard to this value, which is currently unresolved, we have tried to calculate it as a reference value to contrast with past data of Gy-Eq and CBE, since it has been used for a long time in the past. We have included this information because we think that it is desirable to list it in conjunction with the absorbed doses. We added this content in discussion (Line 383-385).
Results:
*Section 3.1: It is interesting to mention that both cell lines accumulate BC-IP in the same way, rather than BPA. This shows one of the issues mentioned in the introduction, related to the differences in BPA uptake between tumors.
Figure 1 C and D: statistical analyses are lacking.
Ans. We added statistical analysis (Fig c,d) (Line 232-234). Additionally, I am very sorry for my mistake that only in C6 at 3h was not significant, and I forgot to write (Line 214 and 229). I replaced a new Figure 1 containing p-value* and error bar in (c,d) for better understanding (Line 225).
* Section 3.2: Please revise redaction and unify redaction with M&M the boron doses used for biodistribution studies. This section was difficult to follow.
Ans. We revised this section. For better understanding, we changed Table 1. The previous table 1 expressed BC-IP concentration and dosage, that was confusing. Thus, we unified only dosages. Please refer to Line 235-256.
Moreover, it would be useful for the reader to explain why the authors chose 500 ugB/ml to explore 3, 6, and 24 hours after injection. Is it based on previous studies?
Ans. As we mentioned earlier, we previously conducted MID-AC/BNCT study at dose of 20mg B/kg. We wanted to confirm boron concentration by time (3, 6, and 24 h) at half dosage, including by way of comparison with our previous data. Please refer to line 241.
Table 1 and figure 2 shows very low boron concentration values in the tumor. Please explain in the text carefully (it was mentioned this in the discussion section).
Ans. We mentioned that boron concentration of the tumor was low (Line 243).
*Section 3.3: Why did you choose 3 h after BC-IP injection if blood has the highest boron concentration at this point? Why you did not choose 6 h after injection that has lower boron concetration in blood?
Ans. We chose 3h because that point was the highest boron concentration in the tumor. We prefer concentration of the tumor to that of blood (Line 258-259).
*Boron concentration values used for dose planning corresponding to BPA group are lacking. How did you prescribe the dose in these BNCT in vivo irradiations? Which is your dose-limiting tissue? Did you evaluate radiotoxicity?
Ans. We are sorry. BPA 12mg B/kg and BC-IP 20mg B/kg was administered to animals. We stated these values in Material and Methods section (line 169-170). In our irradiation experiment, the irradiation time was fixed and limited. We did not prescribe the dose but the neutron dose. Survival intervals were short, but there was no radiotoxicity observed at least during that time (line268-269).
*Section 3.4: “The physical and photon-equivalent doses delivered to the brains and tumors during 263 neutron irradiation experiments were estimated using the CBE factor for each boron car- 264 rier, which was derived from reference data [11]. This estimation also considered the av- 265 erage tumor boron concentrations obtained from in vivo biodistribution studies [16]”
Why the authors did not use the values obtained in the biodistribution studies reported in this article? This comment is related to the comment made before related to Gy-Eq calculations.
Ans. We used reference data [15] of boron concentration only in BPA-BNCT at 2 h group (tumor 17.8, normal brain 4.3 µg B/kg) because we did not conduct BPA biodistribution in this study. We added comment (Line 291-293).
Discussion:
“This study focused on the application of BC-IP, a novel boron carrier, in BNCT for 279 the treatment of high-grade gliomas. Through extensive in vivo studies, including neutron 280 irradiation, we observed promising results, highlighting the potential of BC-IP to enhance 281 the therapeutic potential of BNCT”.
I recommend to erase the word “extensive”, as I the number of animals used in this study is quite low.
Ans. We deleted the word “extensive” (Line 307).
*In the statement “whereas the novel com- 354 pound BC-IP enters the tumor via the EPR effect”.
I recommend to mention “between others” as boron uptake when conjugated with albumin could be also due to an increase in albumin catabolism by the tumor, or other albumin specific transporters like gp60 and SPARC (example Kikuchi et al. 2016 – ref 6).
Ans. Thank you for your suggestion. We added the sentence about catabolism and transporters with new reference (Line 387-390).
- Lin, T.; Zhao, P.; Jiang, Y.; Tang, Y.; Jin, H.; Pan, Z.; He, H.; Yang, V.C.; Huang, Y. Blood-Brain-Barrier-Penetrating Albumin Nanoparticles for Biomimetic Drug Delivery via Albumin-Binding Protein Pathways for Antiglioma Therapy. ACS Nano 2016, 10, 9999-10012, doi:10.1021/acsnano.6b04268.
*Finally, related to your comments on combining BPA + BC-IP and CED protocol, I recommend the authors to analyze and discuss the article by Fukuo et al. 2020 (ref 16) to enrich their future protocols and discuss them in the text.
Ans. We added this sentence mentioned about future work (Line 403-404).
Reviewer 2 Report
The following questions and comments for the authors are provided below.
General Comments:
1. What is the target concentration of BPA versus BC-IP needed to achieve a dosimetric BNCT enhancement effect? The results show that BPA concentrations are much higher than BC-IP
2. Where does the BPA and BC-IP bind onto the receptor(s)of the tumor cells? Do one or both of these bind within the nucleus of the cell where the most tumor damage is likely to occur?
3. The authors claim that “we observed promising results, highlighting the potential of BC-IP to enhance the therapeutic potential of BCNT.” This is based on in vitro experiments showing lower intracellular uptake but a prolonged retention of BC-IP in tumor compared to BPA. Despite the longer BC-IP retention in tumors, in vivo studies showed better BPA results in terms of increased survival times and BNCT estimated dose enhancement. BPA cleared within one hour after exposure when drug administration was terminated. However, tumor uptake of BC-IP is saturated at the start so intracellular boron concentrations are low at the start and completion of irradiation. I am not sure the results support the conclusion that they are “promising”.
Specific Comments:
Page 4, Line 144-145: Is it possible to quantify what is meant by “substantial growth?”
Page 4, Line 145: What were the “predetermined time points?” Please clarify.
Page 4, Line 155: Can you justify why the irradiation group had only 2 rats and the radiation + BC-!P group had 8 rats, while the control and radiation + BPA groups had 5 rats each? Shouldn’t all groups have the same number of rats to eliminate any bias in the results?
Page 4: Paragraph starting at line 170: It is not clear how the authors estimated/measured the dose epithermal, fast neutron, and Gamma emissions . Where specifically was the gold foil placed with respect to the head? Please explain how the authors accounted for dose perturbations caused by having the gold foil in place during irradiations?
Author Response
Dear Reviewer 2,
Thank you very much for reviewing our manuscript and offering valuable advice. We have addressed your comments with point-by-point responses and revised the manuscript accordingly. We believe that the comments have made our manuscript deeper and better than before. All co-authors agreed and we have finished the revision. Please check the history of changes in the manuscript WORD review and responses to the comments below.
General Comments:
What is the target concentration of BPA versus BC-IP needed to achieve a dosimetric BNCT enhancement effect? The results show that BPA concentrations are much higher than BC-IP
Ans. In BPA group, the target boron concentration was generally more than 20 μg B/ml. In BC-IP group, the target concentration was not set in this pilot study, but biological effect of BC-IP was considered to be relatively high because BC-IP/BNCT prolonged survival despite extremely low boron concentration. Thank you for your comment, we added this content in discussion (Line 382-383).
- Where does the BPA and BC-IP bind onto the receptor(s)of the tumor cells? Do one or both of these bind within the nucleus of the cell where the most tumor damage is likely to occur?
Ans. As we describe in discussion section (line 386-390), BPA enters the tumor via LAT-1, whereas BC-IP enters via EPR effect, gp60 or SPARC transporters. It is actually unknown, but taking estimated biological effect into consideration, BC-IP is probably located near nucleus or mitochondria where the most tumor damage occur. We will continue to research.
- The authors claim that “we observed promising results, highlighting the potential of BC-IP to enhance the therapeutic potential of BCNT.” This is based on in vitro experiments showing lower intracellular uptake but a prolonged retention of BC-IP in tumor compared to BPA. Despite the longer BC-IP retention in tumors, in vivo studies showed better BPA results in terms of increased survival times and BNCT estimated dose enhancement. BPA cleared within one hour after exposure when drug administration was terminated. However, tumor uptake of BC-IP is saturated at the start so intracellular boron concentrations are low at the start and completion of irradiation. I am not sure the results support the conclusion that they are “promising”.
Ans. “promising” may be not appropriate. We changed “favorable” (Line 308).
Specific Comments:
Page 4, Line 144-145: Is it possible to quantify what is meant by “substantial growth?”
Ans. This "substantial" was ambiguous, so we deleted this word (Line 157).
Page 4, Line 145: What were the “predetermined time points?” Please clarify.
Ans. This sentence was not enough, so we added the detailed time window (3, 6, or 24h after i.v. BC-IP). Please refer to line 158-159.
Page 4, Line 155: Can you justify why the irradiation group had only 2 rats and the radiation + BC-!P group had 8 rats, while the control and radiation + BPA groups had 5 rats each? Shouldn’t all groups have the same number of rats to eliminate any bias in the results?
Ans, To be honest, n=5 in neutron only irradiation group were originally scheduled to be irradiated, but it was found that 3 rats were not irradiated because they fell outside of the irradiation framework. Therefore, we had to analyze with n=2 in neutron only irradiation group. The survival time for this group was considered reasonable based on our previous studies. The control groups and BPA group were considered adequate at n=5 to reduce the number of animals used. For BC-IP group n=8 was chosen to test the significance of the difference because there are many unknowns, including SD, etc.
Page 4: Paragraph starting at line 170: It is not clear how the authors estimated/measured the dose epithermal, fast neutron, and Gamma emissions . Where specifically was the gold foil placed with respect to the head? Please explain how the authors accounted for dose perturbations caused by having the gold foil in place during irradiations?
Ans. The gold foil (50μm thick, 3mm diameter) was attached to the surface of the heads. Because the gold foil is very thin, the dose perturbation effect from the gold foil was calculated to be negligible. Please refer to line 190-192.
Round 2
Reviewer 1 Report
Thank you to the authors for clearly explained each point of the revision.
I have just a couple of comments:
Please change each time you mentioned BPA mediated by BNCT or MIDAC medited by BNCT for: " BNCT mediated by BPA" or "BNCT mediated by MIDAC". My mistake, sorry.
Could you please check the phrase: Borofalan interview form. Is it correct?
The boron compound scheme I suggest to be Figure 1.
Could you please add more details in the M&M regarding to irradiation conditions? I understood correctly that you fixed neutron dose for all irradiation groups? Thank you.
Author Response
Thank you for reviewing our manuscript and your valuable comments. We really appreciate your high regard. Please refer to our reply below.
Please change each time you mentioned BPA mediated by BNCT or MIDAC medited by BNCT for: " BNCT mediated by BPA" or "BNCT mediated by MIDAC". My mistake, sorry.
Ans. Thank you for pointing. We corrected line 30, 42, 56, 63, 67.
Could you please check the phrase: Borofalan interview form. Is it correct?
Ans. The new reference was added, and other references were renumbered to reflect this change.
[13] Kondo, N.; Hirano, F.; Temma, T. Evaluation of 3-Borono-l-Phenylalanine as a Water-Soluble Boron Neutron Capture Therapy Agent. Pharmaceutics 2022, 14, doi:10.3390/pharmaceutics14051106.
The boron compound scheme I suggest to be Figure 1.
Ans. Thank you for your suggestion. We present compound scheme as Fig 1 (line 96-97). Other figures were renumbered to reflect this change.
Could you please add more details in the M&M regarding to irradiation conditions? I understood correctly that you fixed neutron dose for all irradiation groups? Thank you.
Ans. we mentioned that fixed neutron dose was irradiated for all irradiation groups. Also, we corrected more precise neutron flux (line172-174) and added gamma-ray dose measurement in order to help understand irradiation conditions (line 188-190).
Reviewer 2 Report
My comments were adequately addressed.
Author Response
Thank you for reviewing our manuscript and your valuable comments. We really appreciate your high regard.